# Itraconazole resistance in *Madurella fahalii* linked to a distinct homolog of the gene encoding cytochrome P450 14-α sterol demethylase (CYP51)

**Isato Yoshioka[1,2], Ahmed Hassan Fahal[3], Satoshi Kaneko[4,5], Wei Cao[2], Takashi Yaguchi[1]\***

**1** Medical Mycology Research Center, Chiba University, Chiba, Chiba, Japan, **2** Research Institute for Science and Engineering, Waseda University, Shinjuku-ku, Tokyo, Japan, **3** Mycetoma Research Centre, University of Khartoum, Khartoum, Sudan, **4** School of Tropical Medicine and Global Health, Nagasaki University, Nagasaki, Japan, **5** Department of Ecoepidemiology, Institute of Tropical Medicine (NEKKEN), Nagasaki University, Nagasaki, Japan

\* yaguchi@chiba-u.jp

## Abstract

### Background

Mycetoma is a deep fungal infection caused by several microorganisms, with *Madurella mycetomatis* being the most common causative agent. Another related species, *Madurella fahalii*, is also known to cause eumycetoma. However, unlike *M. mycetomatis*, *M. fahalii* exhibits resistance to itraconazole, the standard treatment for eumycetoma, and the underlying cause of this resistance remains unknown. Therefore, understanding the mechanism of this resistance is critical for developing more effective therapies.

### Principal Findings

Using the high-quality draft genome sequence of *Madurella fahalii* IFM 68171, we identified two copies of the gene encoding cytochrome P450 14-α sterol demethylase (CYP51), the target enzyme of itraconazole. These include a gene conserved among *Madurella* species (*Mfcyp51A1*) and a *M. fahalii*-specific gene (*Mfcyp51A2*). Both genes are actively transcribed in *M. fahalii* and are upregulated in response to itraconazole. Furthermore, heterologous expression in *Saccharomyces cerevisiae* revealed that transformants carrying the *Mfcyp51A2* gene exhibited reduced susceptibility to itraconazole compared to those with *Mfcyp51A1.*

### Conclusion

We demonstrated that itraconazole resistance in *M. fahalii* may be attributed to the presence of an additional CYP51 gene. This study represents the first report on the physiological characteristics of *Madurella* species using genetic engineering techniques.

**Data availability statement:** All relevant data are within the manuscript and its Supporting information files.

**Funding:** This study was supported by AMED under Grant Number JP21jm0510005 in collaboration with FY 2021 International Collaborative Research Program for Tackling the NTDs ( Neglected Tropical Diseases) Challenges in African Countries to SK. This study was also supported by the National Bio-Resource Project, Japan, NBRP_028 to TY. The funder had no role in study design, data collection and analysis, decision to publish, or preparation of the manuscript.

**Competing interests:** The authors have declared that no competing interests exist.

## Author summary

*Madurella fahalii*, a species closely related to *M. mycetomatis*, the most common cause of eumycetoma, exhibits resistance to itraconazole, the standard treatment for this neglected tropical disease. The underlying mechanism of this resistance remains unclear. Understanding it is essential for improving treatment options for *M. fahalii* infections. In this study, using genetic engineering techniques and a high-quality genome sequence of *M. fahalii*, we identified an additional gene associated with itraconazole resistance, which is absent in *M. mycetomatis*. This discovery could pave the way for more effective treatment strategies for eumycetoma caused by *M. fahalii* in the future.

## Introduction

Mycetoma is a neglected tropical disease with profound medical and socioeconomic consequences, affecting individuals and communities in endemic regions worldwide [1]. This chronic, progressive infection, caused by various microorganisms, can lead to severe deformities, disabilities, and, if left untreated, even death. Mycetoma is classified into two types based on the causative agent: eumycetoma, caused by fungal infections, and actinomycetoma, resulting from bacterial infections [2,3]. Among the numerous pathogens responsible for eumycetoma, *Madurella mycetomatis* is the most frequently isolated species [2,3]. However, another important species, *Madurella fahalii*, also plays a critical role in causing eumycetoma [4]. It is of particular concern due to its resistance to itraconazole, the primary drug used in treating this condition [4,5].

The study of azole resistance in fungal pathogens has been well-documented in species such as *Aspergillus*, *Candida*, and *Cryptococcus* [6]. This knowledge has facilitated the development of diagnostic tools and effective treatment strategies. Azole compounds, such as itraconazole, target cytochrome P450 14-α sterol demethylase (CYP51), an enzyme essential for fungal cell membrane synthesis. Resistance to these antifungal agents typically arises from mutations in the *cyp51A* (ERG11) gene, which encodes CYP51. These mutations decrease the efficacy of azoles by altering the binding affinity of the drug to its target. Numerous studies have highlighted this mechanism as a primary contributor to azole resistance in various fungi [7–9]

Genetic engineering techniques have been widely applied in fungal research to investigate resistance mechanisms. One commonly used model organism is *Saccharomyces cerevisiae*, which has proven valuable for studying the molecular basis of drug resistance. These techniques have also been employed in research on pathogenic fungi, aiding in the understanding of how resistance develops and how it can be counteracted [10–12]. However, despite the success of genetic engineering in studying drug resistance in other fungal species, no research has yet applied these methods to *Madurella* species, including *M. mycetomatis* and *M. fahalii*. This lack of genetic studies represents a significant gap in mycetoma research, particularly when compared to progress made in understanding other fungal pathogens.

In this study, we aimed to address this gap by investigating the mechanisms of itraconazole resistance in *M. fahalii* through genome sequencing and genetic engineering approaches. Using *S. cerevisiae* as a model system, we examined how *M. fahalii* develops resistance to itraconazole, and how these insights can improve diagnosis and treatment strategies for eumycetoma caused by this species. Our findings not only provide new insights into itraconazole resistance in *M. fahalii* but also highlight the potential of molecular techniques to advance the study of mycetoma. These results underscore the importance of integrating genetic

engineering into mycetoma research, offering a powerful tool for understanding drug efficacy and resistance in neglected fungal diseases.

## Materials and methods

### Strains and cultivation conditions

*Escherichia coli* DH-5α was used for gene cloning and plasmid maintenance. The *E. coli* strain was cultured in LB medium with the addition of 100 mg/L of ampicillin to maintain a plasmid. *M. mycetomatis* IFM 46458, *M. fahalii* IFM 68171 (preserved as MRC No.13 at the Mycetoma Research Center, Khartoum, Sudan), IFM 68170 (MRC No.9) and IFM 68242 (MRC No. 25) were grown and maintained on Sabouraud medium (FUJIFILM Wako Pure Chemical Corporation, Osaka, Japan) containing 1.5% (w/v) agar. For itraconazole resistance assays, Sabouraud agar medium with varying concentrations of itraconazole (0, 0.008, 0.032, 0.125, and 2 mg/L) was used. For transcriptional analysis, Sabouraud liquid medium with or without 2 mg/L of itraconazole was employed. These *Madurella* strains were preserved at the Medical Mycology Research Center, Chiba University, Japan, through the National Biology Resource Project (NBRP). A stock solution of itraconazole (200 mg/L) was prepared by dissolving it in dimethyl sulfoxide (DMSO), and the medium was supplemented to achieve a final DMSO concentration of 1% (v/v).

$S. cerevisiae$ BY4741 (MATa, his3Δ1, leu2Δ0, met15Δ0, ura3Δ0) was obtained from the NBRP-Yeast Collection (deposited as BY23849, https://yeast.nig.ac.jp/yeast/top.xhtml) and maintained in YPD medium. For heterologous expression of the *cyp51A* gene from *Madurella* strains, *S. cerevisiae* TRE11-4741 (derived from BY4741; $ERG11:LEU2$-$P_{CMV}$-$tTA$-$T_{ADH1}$-$tetO_7$-$P_{CYC1}$-$UAS$-$ERG11$) was generated in this study and used as the host strain, as described below. For yeast transformation and maintenance of transformants, SD-Leu broth (Takara Bio, Shiga, Japan) or SD-Ura medium was used, with 2% (w/v) agar added as necessary. SD-Ura was prepared by combining YNB w/ ammonium sulfate (MP Biomedicals, Santa Ana, CA, USA), CSM-URA (MP Biomedicals), and 2% (w/v) glucose. For the cultivation of yeast strains expressing *cyp51A* heterologously, SG-Ura medium was used, in which 2% (w/v) glucose in SD-Ura was replaced with 2% (w/v) galactose. To knockdown (repress) the endogenous *ERG11* gene in the TRE11-4741 strain, 10 mg/L of doxycycline hydrochloride (FUJIFILM Wako Pure Chemical Corporation) was added to the medium.

### Itraconazole resistance test of *Madurella* strains

Itraconazole resistance was assessed following the method described by du Pré *et al.*, with slight modifications [13]. Briefly, mycelial samples from agar plates were cut using the back of a pipette tip and inoculated onto Sabouraud agar medium containing varying concentrations of itraconazole (ranging from 0 to 2 mg/L). The cultures were incubated at 37°C for three weeks.

### Preparation of fungal genomic DNA

For genome sequencing of *M. fahalii* IFM 68171, 5-7 pieces of 5 mm² mycelial samples were transferred to 1 mL of Sabouraud medium in a microtube, then homogenized by vigorous agitation with an inoculating needle and vortexing. The suspension was subsequently inoculated into 50 mL of Sabouraud medium and cultured at 37°C for three days. The mycelia were harvested by filtration using Miracloth (Merck, Darmstadt, Germany) and ground into a fine powder with liquid nitrogen. Genomic DNA was extracted from the mycelial powder using a phenol-chloroform method and purified with a Genomic-tip 100/G column (Qiagen, Hilden, Germany) [14]. For *M. mycetomatis* IFM 46458, *M. fahalii* IFM 68170 and IFM 68242, genomic DNA extraction was performed as previously described [15].

For *S. cerevisiae*, overnight cultures grown in YPD medium were collected by centrifugation and disrupted in TE buffer using the MagNA Lyser. DNA extraction was then completed using the Maxwell RSC Cultured Cells DNA Kit (Promega, Madison, WI, USA).

## Genome sequencing

The genomic sequencing was conducted by Genome-Lead Co., Ltd. (Kagawa, Japan). For short-read sequencing, a DNA library was prepared using the NovaSeq 6000 SP Reagent Kit v1.5 (Illumina, San Diego, CA, USA), and paired-end (PE) short reads were generated using the Illumina NovaSeq 6000 platform. For long-read sequencing, genomic DNA treated with Short Read Eliminator XS (Circulomics, Baltimore, MD, USA) was used to prepare a DNA library using the Ligation Sequencing Kit V14 (SQK-LSK114, Oxford Nanopore Technologies (ONT), Cambridge, UK). Long reads were sequenced using the PromethION platform (ONT) equipped with an R10.4.1 flow cell (FLO-PRO114M, ONT).

## Genome assembly and annotation

PE short reads were trimmed and filtered by fastp v.0.23.4 [16] with a minimum length of 40 bp and quality score of ≥10. Additionally, ONT long reads were trimmed with Porechop v.0.2.4 (https://github.com/rrwick/Porechop) using default parameters and filtered using NanoFilt v.2.8 [17] by length (≥1000 bp) and quality (≥10). The mitochondrial genome was then assembled from the filtered PE reads using GetOrganelle v.1.7.7 [18] by setting the target organelle as fungal mitochondria (fungus_mt). Reads from both PE and ONT that were not mapped to the mitochondrial genome were recovered using minimap2 v.2.26 [19] and samtools v1.17 [20]. Genome assembly was conducted using the recovered ONT reads with NECAT v.0.0.1 [21] with a default parameter by setting the genome size as 39 Mb. The corrected and trimmed ONT reads generated by NECAT were also assembled using Flye v.2.9.2 [22], followed by combining these assemblies by Quickmerge [23]. The resultant assembly was polished by Medaka v1.9.1 (https://github.com/nanoporetech/medaka) with the long reads and by NextPolish v1.4.1 [24] with the short reads.

The draft genome sequence was masked using RepeatModeler v.2.0.5 and RepeatMasker v.4.1.5 (https://www.repeatmasker.org/) with default parameters. Genome annotation was performed using BRAKER2 v.2.1.6 and AUGUSTUS v.3.4.0 [25,26] with the flag --fungus using odb10_fungi protein sequences from OrthoDB (https://v100.orthodb.org/download/odb10_fungi_fasta.tar.gz) as protein hints. Functional annotation of protein-coding genes was carried out using eggNOG mapper [27], antiSMASH 7.0 [28], Interproscan 5.68-100 [29], Phobius v.1.01 [30] and SignalP 4.1 [31]. tRNAs and rRNAs were annotated by tRNAscan-SE v.2.0.12 [32] and barrnap v.0.9 (https://github.com/tseemann/barrnap), respectively. These annotation results were integrated using the Funannotate v.1.8.15 pipeline (https://github.com/nextgenusfs/funannotate).

## RNA isolation, cDNA preparation and transcriptional analysis

*M. fahalii* IFM 68171 was pre-cultured following the same method used for genomic DNA extraction. Mycelia were collected by filtration using Miracloth, and 100 mg of wet mycelia were inoculated into 20 mL of Sabouraud medium with or without 2 mg/mL itraconazole. *M. mycetomatis* IFM 46458 was cultured only in the medium without itraconazole. After 24 h of incubation, the mycelia were recovered using Miracloth and ground in liquid nitrogen. Total RNA was then extracted from the powdered mycelia using the NucleoSpin RNA Plant and Fungi kit (Takara) and cDNA was synthesized using PrimeScript RT reagent Kit with gDNA Eraser (Perfect Real Time) (Takara), following the manufacturer's protocol. Quantitative PCR

(qPCR) was employed to quantify the transcriptional levels of each gene using TB Green Premix Ex Taq II (Tli RNaseH Plus) (Takara), in accordance with the manufacturer's instructions. A primer pair, Pr1-Pr2, was used for PCR amplification to detect the entire ORF from genomic DNA and the corresponding cDNA regions. The transcription levels of *cyp51A* were measured using primer pairs Pr3-Pr4 or Pr5-Pr6, and they were normalized against actin and tubulin, amplified with primer pairs Pr7-Pr8 and Pr9-Pr10, respectively. The primer sequences used in this study are listed in S1 Table.

## Construction of plasmids and transformation of yeast

To create a host strain for expressing *cyp51A* genes, a doxycycline-repressible expression cassette for *ERG11* (*cyp51A* orthologue in *S. cerevisiae*) was constructed [33], following the approach by Groeneveld *et al*. [34]. The 5′-flanking region and open reading frame (ORF) of *ERG11* were separately amplified from the genomic DNA of strain BY4741 via PCR, using primer pairs Pr11-Pr12 and Pr13-Pr14, respectively. Additionally, a *LEU2* fragment was obtained by PCR using primers Pr15-Pr16 from BYP5029 (pGG115 [35]), which was sourced from the NBRP-Yeast collection. In parallel, the doxycycline-regulatable elements ($P_{CMV}$-tTA-$T_{ADH1}$-tetO$_7$-$P_{CYC1}$-UAS) were amplified using primers Pr17-Pr18 from plasmid BYP7139 (NBRP-Yeast collection), derived from pCM190 [36]. All PCR fragments were assembled via cloning into *EcoRI*-digested pUC19 using the In-Fusion HD Cloning Kit (Takara) according to the manufacturer's instructions, to yield tet-*ERG11* cassette consisting of the flanking region of *ERG11*, *LEU2* gene, the doxycycline-regulatable elements and the ORF of *ERG11*. Subsequently, tet-*ERG11* cassette was then amplified from the resulting plasmid using primers Pr11-Pr14 and introduced into strain BY4741 via the Fast Yeast Transformation Kit (G-Biosciences, St. Louis, MO, USA). The transformation mixture was cultured for three days on SD-Leu agar at 30°C, followed by subculturing under the same conditions.

To evaluate the *cyp51A* genes from *Madurella* species, their cDNAs were amplified using primer pairs Pr19-Pr21, Pr20-Pr21, and Pr22-Pr23. These PCR fragments were cloned into *EcoRI*-digested yeast expression vector pYES2 (Thermo Fisher Scientific, MA, USA) using the In-Fusion HD Cloning Kit. Additionally, the *ERG11* gene was amplified from the genomic DNA of *S. cerevisiae* using primers Pr24-Pr25 and cloned into pYES2 in the same manner. These plasmids were subsequently introduced into strain TRE11-4741 using the Fast Yeast Transformation Kit, and the transformants were grown on SD-Ura agar. Similarly, transformants harboring the empty pYES2 plasmid were prepared as a negative control.

## Phenotypic assay and drug resistance testing of yeast transformants

To assess the ability of *cyp51A* genes to complement *ERG11* knockdown, yeast strains harboring pYES2 derivatives were cultured in SG-Ura liquid medium at 30°C overnight to induce *cyp51A* expression via galactose [37]. The culture broth was serially diluted 10-fold, from $2\times10^7$ to $2\times10^4$ cells/mL using SG-Ura medium, and 5 μL of each dilution was spotted onto SG-Ura agar with or without doxycycline. The plates were then incubated for three days at 30°C.

Drug resistance testing was performed using a modified broth microdilution method based on the protocol by Martel *et al*. with slight modifications [10]. Briefly, yeast cells cultured overnight in SG-Ura medium were adjusted to a concentration of $1\times10^4$ cells/mL in SG medium containing 10 mg/L of doxycycline and inoculated onto the Dried Plate for Antifungal Susceptibility Testing of Yeasts (Eiken Chemicals, Tokyo, Japan). The microtiter plates were incubated at 30°C for three days, after which the optical density at 630 nm for each well was measured using an iMark Microplate Reader (Bio-Rad Laboratories Inc., Hercules, CA,

USA). The minimal inhibitory concentration (MIC) was determined as the concentration that inhibited yeast growth by 80%

## Homology modeling and molecular docking

The 3D models of MFCYP51A1 and MFCYP51A2 were generated using AlphaFold2 v.2.3.2 with a default parameter [38]. These models were refined using openMM v.8.2.0 to minimize the local energy [39]. The 3D structure of itraconazole was retrieved from PubChem (ID: 55283), and used for docking simulation by AutoDock Vina v.1.1.2 [40]. The optimal structure of protein-ligand complex was chosen based on the crystal structure of 14-alpha sterol demethylase derived from *A. fumigatus* Af293, which was retrieved from PDB (ID: 6CR2).

Molecular dynamics (MD) simulations were conducted using AmberTools v.23.4 [41] and GROMACS v.2024.4 [42]. For the force fields, FF19SB was applied to proteins, TIP3P to water molecules, and GAFF2 to itraconazole. Regarding heme, a cofactor located in the active center of CYP51, quantum mechanical calculations were performed by Gaussian v.16 (Gaussian, Inc., Wallingford CT, USA) using the density functional theory (DFT) at the B3LYP/6-31G* level [43], and the resultant model were subsequently integrated into the MD simulations. The system was equilibrated under in the NVT ensemble for 1 ns, followed by NPT equilibration to stabilize the pressure at 1 bar and temperature at 300 K. Finally, the MD simulations were run for 150 ns, during which the temperature was maintained at 300 K using the V-rescale thermostat, and the pressure at 1 bar using the Parrinello−Rahman barostat. Particle Mesh Ewald (PME) was used to calculate electrostatic interactions, with cutoff distances of 10 Å for Coulombic, electrostatic, and van der Waals interactions. For the visualization and analysis of data, PyMOL v.3.0.0 (http://www.pymol.org/pymol) was used. The total binding energy between CYP51 and itraconazole were calculated using gmx_MMPBSA v.1.6.2 [44].

## Results

### Genome sequencing of *Madurella fahalii* IFM 68171

After trimming and filtering the sequencing reads, we obtained 36,802,641 PE reads (10.9 Gb) and 491,402 ONT reads (4.07 Gb). The PE reads were used for assembling the mitochondrial genome. For the chromosomal genome assembly, 422,878 ONT reads (3.50 Gb, Coverage x87.4) with an average length of 8.27 kb, along with 34,342,271 PE reads (10.2 Gb), were filtered through mitochondrial genome mapping. This process resulted in a chromosomal genome assembly consisting of six scaffolds with a total length of 40,045,822 bp and an N50 value of 25,195,283 bp. Five of the scaffolds contained telomeric sequences at both ends, while the remaining scaffold a telomeric sequence at one end, as shown in S1 Fig. Genome annotation identified 11,334 protein-coding genes, 180 tRNAs, and 79 rRNAs. The BUSCO v.5.5.0 [45] score based on the eukaryote and sordariomycetes databases were 100% and 99.6%, respectively. These results indicated that a high-quality genome assembly of *M. fahalii* IFM 68171 was obtained. Additionally, a 40,079 bp mitochondrial genome was assembled. The genomic and mitochondrial sequences have been deposited in DDBJ/EMBL/GenBank under accession numbers BAAFSV010000001 - BAAFSV010000006 and LC843096, respectively.

### Identification of *cyp51A* gene in *Madurella fahalii*

Genome annotation revealed that *M. fahalii* IFM 68171 possesses two *cyp51A* genes encoding cytochrome P450 14-α sterol demethylase (CYP51), the target enzyme of itraconazole. One of these genes (*Mfcyp51A1*; locus tag MFIFM68171_00517) is conserved in *M. mycetomatis* [46], while the second gene (*Mfcyp51A2*; locus tag MFIFM68171_05904) is specific to *M. fahalii*. The putative coding sequence (CDS) region of *Mfcyp51A2* is illustrated in S2 Fig. In addition,

genomic and RT-PCR analyses, which amplify the entire ORF (CDS) region, revealed that *Mfcyp51A2* was transcribed as shown in Fig 1. Notably, no homologous *Mfcyp51A2* gene exists in the genome of *M. mycetomatis*, although its adjacent gene locus is conserved, as depicted in S3 Fig. Furthermore, *Mfcyp51A2* was successfully amplified from the genomic DNA of other *M. fahalii* strains (IFM 68170 and IFM 68242), while it was not detected in *M. mycetomatis* IFM 46458, as shown in Fig 2. The amino acid sequence of the protein encoded by *Mfcyp51A2* shows 70% similarity to *cyp51A* from *M. mycetomatis* strain mm55 (*Mmcyp51A*; accession number KXX80456.1), which has an identical sequence to strain IFM 46458.

A BLASTP search in the nr-database for Fungi (taxid: 4751) revealed that MFCYP51A2 shows strong similarity to *cyp51A* from other fungal species, as summarized in Table 1. Furthermore, as shown in Fig 3, amino acid alignments between MFCYP51A2 and other CYP51 enzymes derived from *M. mycetomatis*, *Chaetomium globosum*, *Aspergillus fumigatus*, *Candida albicans*, and *S. cerevisiae* indicate that MFCYP51A2 retains critical domains found in fungal CYP51, including six substrate recognition sites (SRS) and three conserved motifs [47].

## Itraconazole resistance of *M. fahalii* IFM 68171

*M. fahalii* IFM 68171 and *M. mycetomatis* IFM 46458 were cultivated on Sabouraud media containing varying concentrations of itraconazole to assess the resistance of strain IFM 68171. As shown in Fig 4, *M. fahalii* exhibited growth even in media containing 2 mg/L of

**(A)**

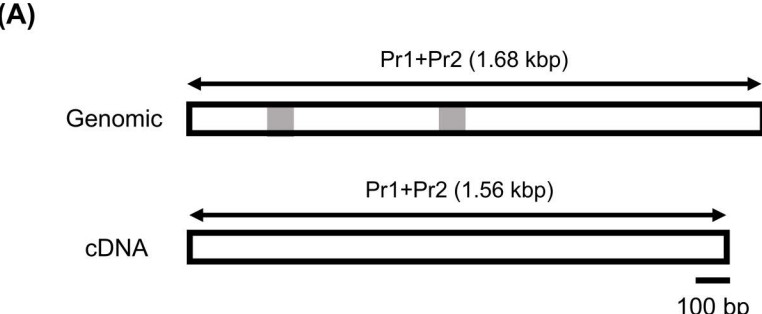

**(B)**

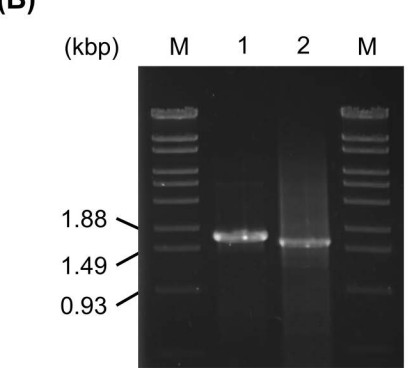

**Fig 1. PCR analysis of the Mfcyp51A2 gene in *M. fahalii* IFM 68171.** (A) Schematic representation of the genomic DNA and cDNA, showing the length of the PCR product amplified using the primer pair Pr1-Pr2. The intron region is depicted in gray. (B) Agarose gel electrophoresis of PCR products. Lanes: M, λ/StyI (Marker 6, Nippon Gene, Toyama, Japan); 1, genomic DNA; 2, cDNA.

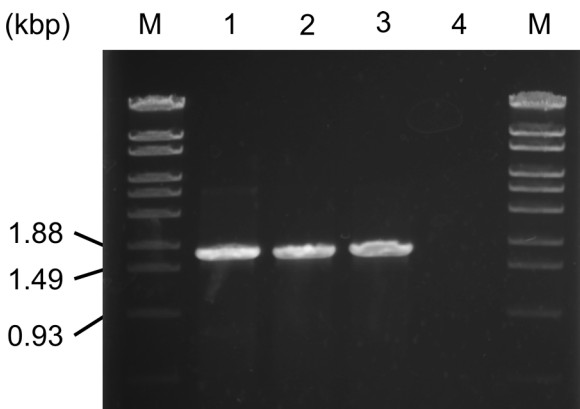

**Fig 2. Detection of *Mfcyp51A2* from the genomic DNAs in *Madurella* strains.** PCR was performed using the primer pair Pr1-Pr2. Lanes: M, λ/StyI (Marker 6, Nippon Gene); 1, *M. fahalii* IFM 68171; 2, *M. fahalii* IFM 68170; 3, *M. fahalii* IFM 68242; 4, *M. mycetomatis* IFM 46458.

**Table 1. BLAST search result using MFCYP51A2 as a query.**

| Annotation | Species | Max Score | Query Cover | Identity | Accession |
|---|---|---|---|---|---|
| 14-alpha sterol demethylase Cyp51A | *Hyaloscypha* sp. | 946 | 99% | 87% | KAH8746640.1 |
| cytochrome P450 | *Xylariales* sp. | 929 | 97% | 86% | KAH8660298.1 |
| eburicol 14-alpha-demethylase | *Valsa mali* | 906 | 99% | 82% | KUI72529.1 |
| eburicol 14-alpha-demethylase | *Valsa mali* var. *pyri* (nom. inval.) | 899 | 96% | 84% | KUI52487.1 |
| hypothetical protein | *Colletotrichum jinshuiense* | 897 | 96% | 86% | WYZ43662.1 |

itraconazole, whereas the growth of *M. mycetomatis* was completely inhibited at a concentration of 0.125 mg/L.

## Itraconazole-induced transcription of *cyp51A* genes

The transcriptional activity of the *Mfcyp51A* genes in *M. fahalii* was analyzed both in the presence and absence of 2 mg/L itraconazole. To estimate the transcriptional levels, two housekeeping genes (actin and tubulin) [13] were used for internal standards. The primer efficiency was determined using cDNA solutions by plotting a standard curve that were serially diluted by 10-fold, as shown in S4 Fig and S2 Table. The results of qPCR was summarized in Table 2. In the absence of itraconazole, the transcriptional level of *Mfcyp51A1* was 1.89 times higher than that of *Mfcyp51A2*. However, upon the addition of itraconazole, transcription levels of *Mfcyp51A1* increased by 1.25- or 1.96-fold while those of *Mfcyp51A2* increased by 2.00- or 3.14-fold.

## Functional analysis of *cyp51A* derived from *Madurella*

To evaluate the role of *cyp51A* genes in itraconazole resistance, we generated a budding yeast strain with the *ERG11* gene replaced under the control of a *tetO$_7$* promoter. The resulting strain, TRE11-4741, was successfully created, and its growth was tightly suppressed by 10 mg/L of doxycycline (DOX), as shown in S5 Fig. Next, complementation assays were conducted using *cyp51A* (*ERG11*) genes derived from *S. cerevisiae*, *M. mycetomatis*, and *M. fahalii*, driven by a galactose-inducible promoter (P$_{GAL1}$). When *Mmcyp51A* and *Mfcyp51A1* were introduced, the growth repression observed in the control strain harboring the empty vector was relieved,

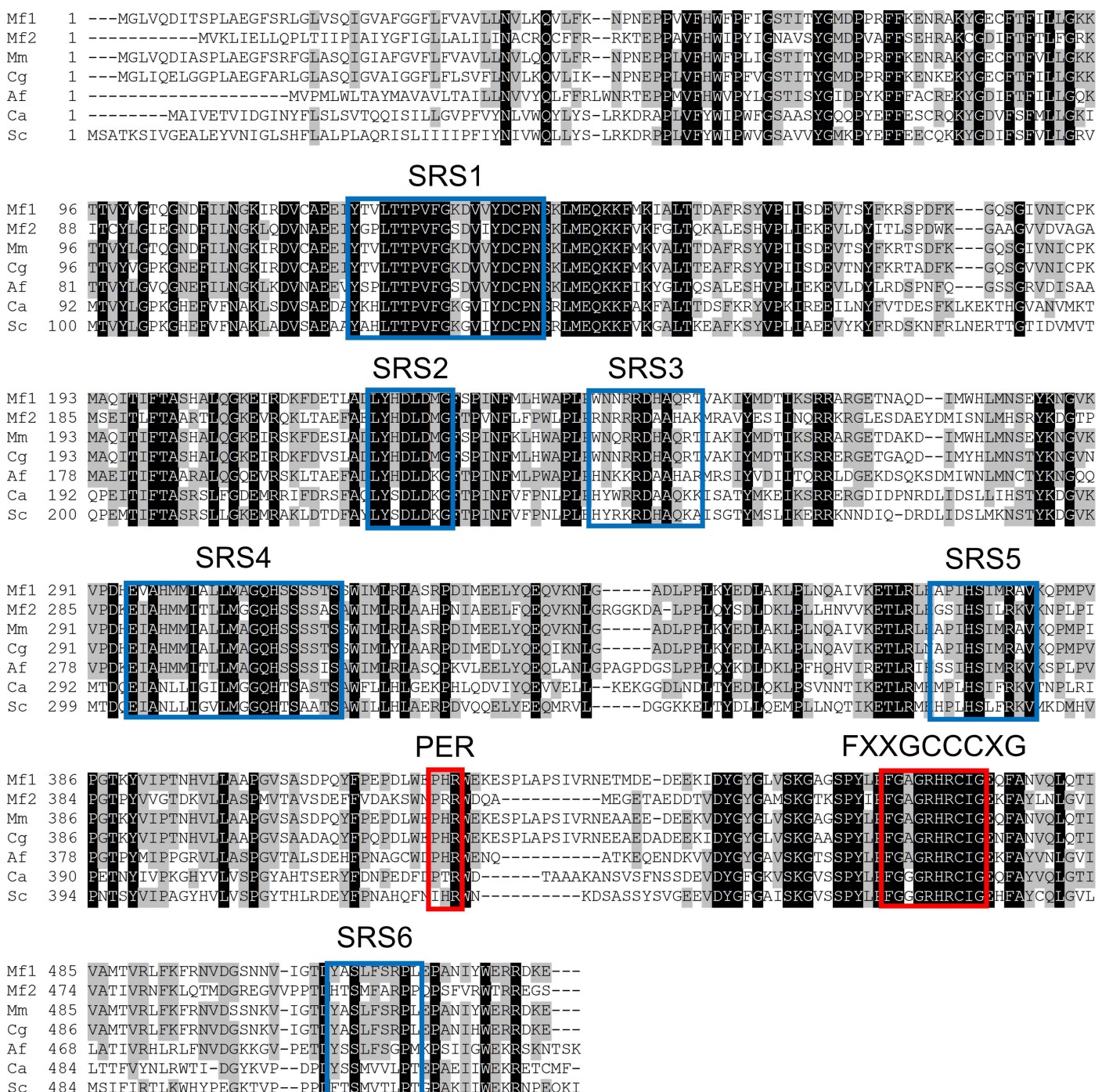

**Fig 3. Amino acids alignment of CYP51 proteins among fungi.** The calculation for alignment and representing the image were performed by Genetyx v.14 (Genetyx Corp. Tokyo Japan). Substrate recognition sites (SRS) and other conservation motifs in fungal CYP51 were highlighted in blue and red boxes, respectively. Abbreviations: Mf1, MFCYP51A1; Mf2, MFCYP51A2; Mm, MMCYP51A; Cg, CYP51 derived from *Chaetomium globosum* (XP_001220873.1); Af, CYP51A derived from *A. fumigatus* (XP_752137.1); Ca, Erg11 derived from *Candida albicans* (XP_716761.1); Sc, Erg11 derived from *S. cerevisiae* (NP_011871.1).

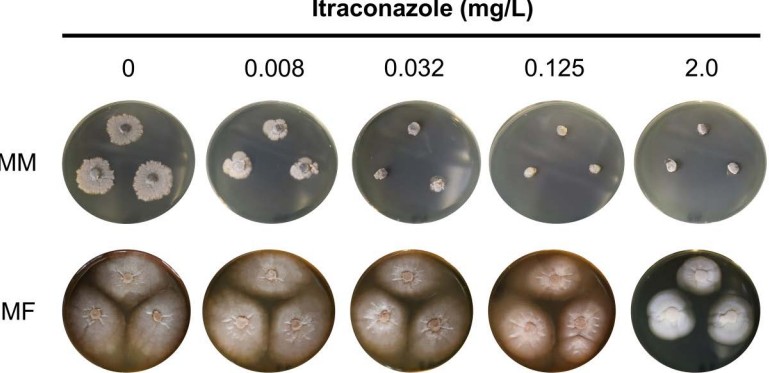

**Fig 4. Growth assay of *Madurella* spp. on Sabouraud medium containing 0, 0.008, 0,032, 0.125 and 2.0 mg/L itraconazole.** MM and MF represent *M. mycetomatis* IFM 46458 and *M. fahalii* IFM 68171, respectively.

**Table 2. Transcriptional level changes of *Mfcyp51A* genes normalized to actin or tubulin gene in the presence of itraconazole.**

| Gene | ΔCt (-azole)[1] | | ΔCt (+azole)[1] | | Relative transcription (= $2^{-\Delta\Delta Ct}$ value) [2] | |
|---|---|---|---|---|---|---|
| | to actin | to tubulin | to actin | to tubulin | to actin | to tubulin |
| *Mfcyp51A1* | 0.06 | -0.75 | -0.26 | -1.72 | 1.25 | 1.96 |
| *Mfcyp51A2* | 0.98 | 0.17 | -0.02 | -1.48 | 2 | 3.14 |

[1]-azole and +azole represent "cultivation under the condition with no additives" and "cultivation under the condition with 2 mg/L itraconazole", respectively.

[2]The ΔΔCt value was calculated by subtracting ΔCt (-azole) from ΔCt (+azole).

comparable to the strain expressing *ERG11*, as shown in Fig 5. Additionally, the strain expressing *Mfcyp51A2* also showed complementation of *ERG11* knockdown, although its growth rate was slightly lower than that of *Mfcyp51A1* (Fig 4). Finally, MIC tests revealed that the strain expressing *Mfcyp51A2* displayed reduced susceptibility to all tested azoles, including fluconazole (16 mg/L), itraconazole (1 mg/L), voriconazole (0.06 mg/L), and miconazole (1 mg/L), compared to strains expressing *Mmcyp51A* and *Mfcyp51A1*, as detailed in Table 3.

## MD simulations of CYP51 proteins and itraconazole

The protein models of MFCYP51A1 and MFCYP51A2 were successfully constructed using AlphaFold2 and refined with OpenMM. The results indicated that the active sites of both enzymes are well-conserved, as shown in Fig 6. Docking simulations demonstrated that itraconazole binds to the active sites of both MFCYP51A1 and MFCYP51A2, as shown in Fig 7. However, the orientation of the nitrogen atom in itraconazole relative to the heme iron differs between the two proteins, as shown in Fig 7. Furthermore, MD simulations demonstrated that the average distance between the heme iron and the nitrogen atom in itraconazole was 2.67 Å for MFCYP51A1 and 3.47 Å for MFCYP51A2, as shown in S6 Fig. The total binding energies for itraconazole of MFCYP51A1 and MFCYP51A2 were -71.38 and -71.69 kcal/mol, respectively.

## Discussion

*M. fahalii* is recognized as a causative agent of mycetoma and has been shown to resist itraconazole, the preferred treatment for fungal mycetoma (eumycetoma) [1,4,48]. In this study, we present a high-quality draft genome sequence of *M. fahalii* IFM 68171 to investigate

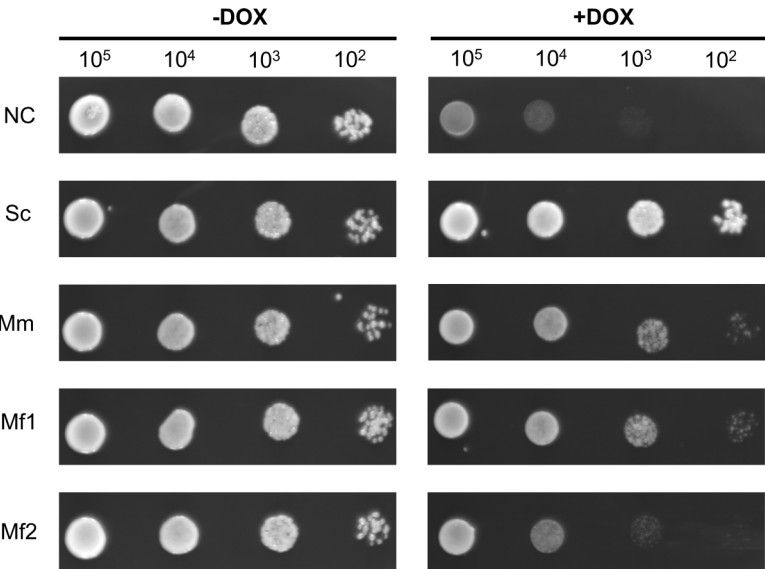

**Fig 5. Growth assay of *S. cerevisiae* strains with DOX-regulated *ERG11* and harboring plasmid expressing *cyp51A* (*ERG11*).** Yeast cells were serially diluted from $10^5$ to $10^5$ by 10-fold and inoculated to media without DOX (-DOX) or with 10 mg/L DOX (+DOX). Abbreviations: NC, pYES2 (empty vector); Sc, *ERG11*; Mm, *Mmcyp51A*; Mf1, *Mfcyp51A1*; Mf2, *Mfcyp51A2*.

**Table 3. MIC values of antifungal drugs in *S. cerevisiae* strains expressing *cyp51A* derived from *Madurella* species.**

| *cyp51A* | MIC (mg/L) | | | | | | | |
|---|---|---|---|---|---|---|---|---|
| | **MCFG** | **CPFG** | **AMPH-B** | **5-FC** | **FLCZ** | **ITCZ** | **VRCZ** | **MCZ** |
| *Mmcyp51A* | 0.12 | 0.25 | 0.5 | 1 | <0.12 | <0.015 | <0.015 | <0.03 |
| *Mfcyp51A1* | 0.12 | 0.25 | 0.5 | 1 | <0.12 | <0.015 | <0.015 | <0.03 |
| *Mfcyp51A2* | 0.12 | 0.25 | 0.5 | 1 | 16 | 1 | 0.06 | 1 |

Abbreviations: MCFG, micafungin; CPFG, caspofungin; AMPH-B, amphotericin B; 5-FC, 5-fluorocytosine; FLCZ, fluconazole; ITCZ, itraconazole; VRCZ, voriconazole; MCZ, miconazole

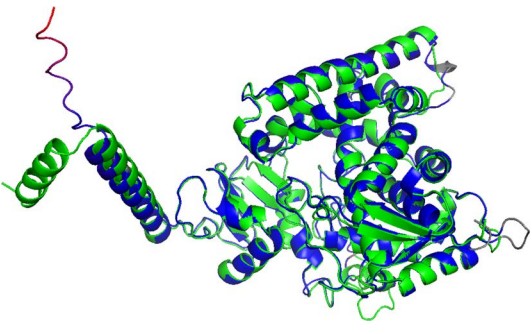

**Fig 6. Cartoon representations of modeled CYP51 proteins derived from *M. fahalii*.** MFCYP51A1 were illustrated in green. The color of MFCYP51A2 changed from blue to red (low to high) based on their RMSD values with the average structure.

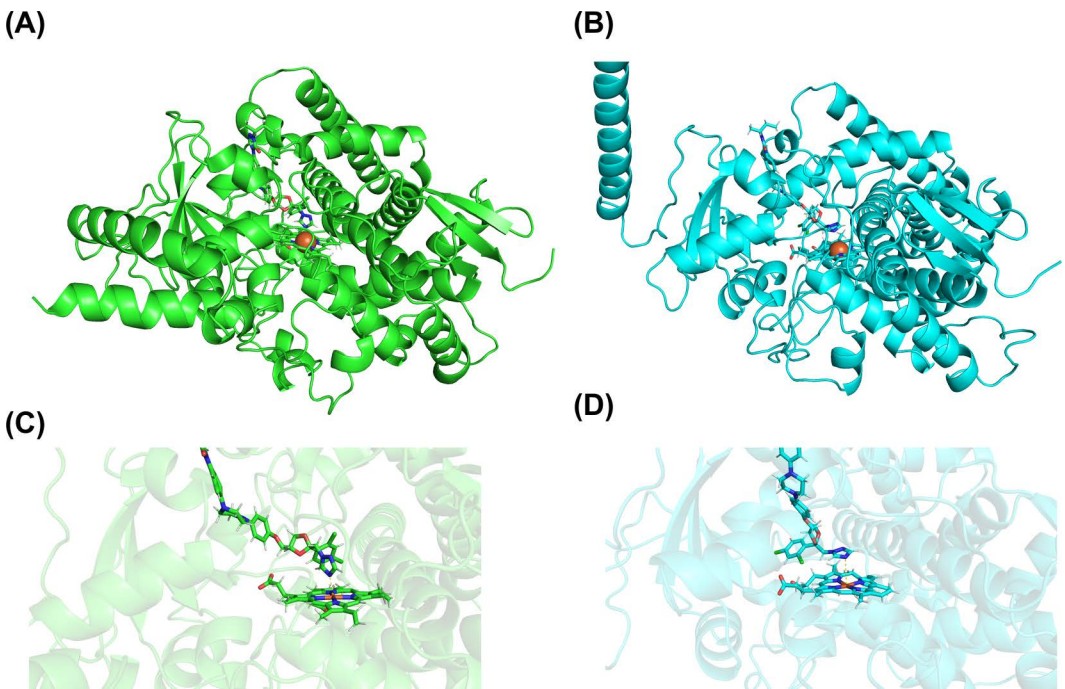

**Fig 7. Representation of the models of MFCYP51A (A, C) and MFCYP51A2 (B, D).** In each panel, itraconazole and heme are illustrated in a stick representation at the upper and bottom part, respectively. (A) and (B) represent the overall structure of CYP51 proteins. The active centers of (A) and (B) were enlarged with their protein backbones translucent to produce (C) and (D), respectively.

the mechanisms underlying its resistance to itraconazole. Genome analysis revealed the presence of two copies of the *cyp51A* gene, encoding cytochrome P450 14-α sterol demethylase (CYP51), the enzyme targeted by itraconazole. This differs from *M. mycetomatis*, which contains only a single copy of the gene (*Mmcyp51A*). Transcriptional analysis showed that both *cyp51A* genes, *Mfcyp51A1* and *Mfcyp51A2*, are transcribed and upregulated in response to itraconazole exposure. Additionally, molecular studies using *S. cerevisiae* as a model system demonstrated that the CYP51 enzyme encoded by *Mfcyp51A2* plays a more significant role in itraconazole resistance compared to *Mfcyp51A1* and *Mmcyp51A*. This marks the first study to directly describe azole resistance in *Madurella* species using genetic engineering techniques.

Both *Mfcyp51A1* and *Mfcyp51A2* contain conserved regions essential for CYP51 function, and both rescued the knockdown of *ERG11* in *S. cerevisiae*. Furthermore, transcriptional upregulation of these genes was observed in response to azole treatment. This suggests that both *cyp51A* genes perform dual roles in ergosterol biosynthesis and in mediating responses to azole-induced stress, consistent with findings in other fungi [49,50]. Notably, it was suggested that *Mfcyp51A2* plays a major role in azole stress response because it shows a significantly stronger induction compared to *Mfcyp51A1*, as shown in Table 2. Regarding protein modeling, the result of MD simulation demonstrated that the distance between heme iron and the nitrogen atom of itraconazole in MFCYP51A2 was greater than MFCYP51A1, while the total binding energies of the two enzymes are nearly equal (Fig 7). Previous studies reported that the iron-nitrogen distance is an important factor to form the Fe-N coordination bond essential for azole inhibition [51,52]. Thus, our result suggests that MFCYP51A2 exhibits a weak interaction with itraconazole in comparison with MFCYP51A1, which is consistent with the differences in itraconazole susceptibility between *S. cerevisiae* transformants expressing

*cyp51A* genes. Taken together, our results indicate that *Mfcyp51A2* is the primary contributor to itraconazole resistance.

A recent study reported the draft genome sequence of *M. fahalii* CBS 129176, another strain exhibiting itraconazole resistance [53]. This strain harbored a mutation (I152V) in *Mfcyp51A1*, near the azole-binding site. However, our findings suggest that this mutation does not contribute to resistance, because the yeast transformant expressing this mutated version of *Mfcyp51A1* displayed MIC values similar to those of *Mmcyp51A*. On the other hand, the homologous *Mfcyp51A2* gene with 100% identity was present in *M. fahalii* CBS 129176 (accession no. JAPYLN010000001.1, 1409341 to 1411021), reinforcing the hypothesis that *Mfcyp51A2* plays a key role in itraconazole resistance. Moreover, we have confirmed that *Mfcyp51A2* is conserved among *M. fahalii* strains (Fig 2), suggesting that *Mfcyp51A2* as well as *Mfcyp51A1* could be important targets for discussing azole resistance in this species, such as the variability in azole resistance in clinical isolates and molecular genetic analysis of the existing resistant strains.

This study identifies *Mfcyp51A2* as a potential target for itraconazole resistance in *M. fahalii.* Previous studies on azole resistance in fungi arising from the mutation of CYP51 have focused on enzyme activity [54]. Thus, further research based on the enzymatic properties of fungal CYP51 will be required to analyze and evaluate *Mfcyp51A2* to support the development of new drugs and treatment strategies. Another potential mechanism of resistance involves the overexpression of genes encoding azole efflux pumps [55–57]. Therefore, it is crucial to investigate the role of efflux pump genes in *M. fahalii* to understand its resistance to azoles. Additionally, regulatory genes involved in drug resistance should be studied to overcome this challenge [58–60]. Future *in vivo* studies on *Madurella* species will be essential for improving treatment options. Although transformation methods have been developed for *M. mycetomatis* [13], similar approaches are needed to explore drug resistance mechanisms in other *Madurella* species. This study provides the first insights into these mechanisms.

In conclusion, this study successfully identified *Mfcyp51A2* as a key gene contributing to itraconazole resistance in *M. fahalii* through genomic and genetic engineering analyses using *S. cerevisiae*. These findings highlight the potential of molecular techniques in uncovering drug resistance mechanisms in neglected fungal pathogens like *Madurella* species.

## Supporting information

**S1 Table.  Primers used in this study.**
(XLSX)

**S2 Table.  The properties of standard curves of qPCR primers.**
(XLSX)

**S1 Fig.  The genomic scaffold of *Madurella fahalii* IFM 68171.** The figure was generated using Tapestry (https://github.com/johnomics/tapestry) by mapping long reads, which were used as input for Flye assembly. Regions corresponding to telomeric sequences (CCCTAA/ TTAGGG) are indicated in red and the opacity represents the number of telomeric repeats. (TIF)

**S2 Fig.  Nucleotides and amino acids sequence of *Mfcyp51A2*.** Introns were highlighted with gray, and encoded amino acids were represented under the corresponding nucleotide sequences. (TIF)

**S3 Fig.  Comparison of genomic region including *cyp51A2* and its flanking genes in *M. fahalii* IFM 68171 (accession no.: BAAFSV010000003.1) with that of *M. mycetomatis***

**m55 (accession no.: LCTW02000001.1).** The amino acids alignment and the generation of the image was performed by Clinker (https://github.com/gamcil/clinker). The homologous protein-coding gene models were drawn as the arrows in the same color and their similarity in amino acids were labeled between them.
(TIF)

**S4 Fig. The standard curves of qPCR primers used in this study to amplify (A) actin, (B) tubulin, (C) *Mfcyp51A1* and (D) *Mfcyp51A2*.** cDNA solutions were serially diluted 10-fold and were used as input. The x-axis represents the log10 of the DNA dilution factor, while the y-axis represents the Ct values.
(TIF)

**S5 Fig. Growth assay of *S. cerevisiae* strains BY4741 and TRE11-4741, a tranformant with doxycycline (DOX)-regulable *ERG11*.** (A) Schematic representation of *ERG11* locus in strains BY4741 and TRE11-4741. In the genome of TRE11-4741, *LEU2* gene, $P_{CMV}$ -*tTA*-$T_{ADH1}$ cassette (*tetR*) and $tetO_7$ -$P_{CYC1}$ -*UAS* cassette ($P_{tet}$) were integrated into the upstream region of *ERG11* ORF in comparison with that of strain BY4741.The genomic region and vector backbone were represented as solid and dotted lines, respectively. The expression of *ERG11* by strain TRE11-4741 is repressed by the addition of DOX. (B) Restricted growth of strain TRE11-4741 by the addition of DOX. Yeast cells were serially diluted from $10^5$ to $10^2$ by 10-fold and cultivated on YPD media without DOX (-DOX) or with 10 mg/L DOX (+DOX) for 2 days.
(TIF)

**S6 Fig. Time-course analysis of the distance between the nitrogen atom in itraconazole and the heme iron during MD simulations of CYP51 proteins.** Green and blue plots represent the data for MFCYP51A1 and MFCYP51A2, respectively.
(TIF)

## Author contributions

**Conceptualization:** Ahmed Hassan Fahal, Satoshi Kaneko, Takashi Yaguchi.

**Data curation:** Isato Yoshioka, Wei Cao, Takashi Yaguchi.

**Formal analysis:** Isato Yoshioka.

**Funding acquisition:** Ahmed Hassan Fahal, Satoshi Kaneko.

**Investigation:** Isato Yoshioka, Wei Cao, Takashi Yaguchi.

**Methodology:** Isato Yoshioka, Wei Cao, Takashi Yaguchi.

**Project administration:** Ahmed Hassan Fahal, Satoshi Kaneko.

**Resources:** Ahmed Hassan Fahal, Takashi Yaguchi.

**Software:** Isato Yoshioka, Wei Cao.

**Supervision:** Ahmed Hassan Fahal, Satoshi Kaneko, Takashi Yaguchi.

**Visualization:** Isato Yoshioka, Wei Cao.

**Writing – original draft:** Isato Yoshioka.

**Writing – review & editing:** Ahmed Hassan Fahal, Satoshi Kaneko, Takashi Yaguchi.

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
