## [Decision Letter · Decision Letter 0]

5 Dec 2024

PNTD-D-24-01441

Itraconazole Resistance in Madurella fahalii Linked to a Distinct Homolog of the Gene Encoding Cytochrome P450 14-α Sterol Demethylase (CYP51)

Dear Dr. Yaguchi,

Thank you for submitting your manuscript to PLOS Neglected Tropical Diseases. After careful consideration, we feel that it has merit but does not fully meet PLOS Neglected Tropical Diseases's publication criteria as it currently stands. Therefore, we invite you to submit a revised version of the manuscript that addresses the points raised during the review process.

Please submit your revised manuscript within 60 days Feb 03 2025 11:59PM. If you will need more time than this to complete your revisions, please reply to this message or contact the journal office at plosntds@plos.org. Please include the following items when submitting your revised manuscript:

We look forward to receiving your revised manuscript.

Kind regards,

Max Carlos Ramírez-Soto, BSc, MPH, PhD, FRSPH

Academic Editor

Marcio Rodrigues

Section Editor

Shaden Kamhawi

co-Editor-in-Chief

Paul Brindley

co-Editor-in-Chief

**Additional Editor Comments :**

I suggest including a paragraph describing and discussing the limitations of your study.

**Journal Requirements:**

At this stage, the following Authors/Authors require contributions: Isato Yoshioka, Ahmed Hassan Fahal, Satoshi Kaneko, and Takashi Yaguchi. Please ensure that the full contributions of each author are acknowledged in the "Add/Edit/Remove Authors" section of our submission form.

2) We noticed that you used the phrase 'not shown' in the manuscript. We do not allow these references, as the PLOS data access policy requires that all data be either published with the manuscript or made available in a publicly accessible database. Please amend the supplementary material to include the referenced data or remove the references.

- ® on pages: 7, 10, 12, and 13

- TM on page: 10.

4) We have noticed that Figure S3 is labeled as Figure 4 . Please check the label of the figure and amend it accordingly. 

5) We have noticed that you have uploaded Supporting Information files, but you have not included a list of legends. Please add a full list of legends for your Supporting Information files after the references list.

6) We note that your Data Availability Statement is currently as follows: "All relevant data are within the manuscript and its Supporting Information files." Please confirm at this time whether or not your submission contains all raw data required to replicate the results of your study. Authors must share the “minimal data set” for their submission. PLOS defines the minimal data set to consist of the data required to replicate all study findings reported in the article, as well as related metadata and methods (https://journals.plos.org/plosone/s/data-availability#loc-minimal-data-set-definition).

7) Please amend your detailed Financial Disclosure statement. This is published with the article. It must therefore be completed in full sentences and contain the exact wording you wish to be published.

2) State what role the funders took in the study. If the funders had no role in your study, please state: "The funders had no role in study design, data collection and analysis, decision to publish, or preparation of the manuscript.".

If you did not receive any funding for this study, please simply state: The authors received no specific funding for this work.

**Reviewers' Comments:**

Reviewer's Responses to Questions

**Key Review Criteria Required for Acceptance?**

**Methods**

-Are the objectives of the study clearly articulated with a clear testable hypothesis stated?

-Is the study design appropriate to address the stated objectives?

-Is the population clearly described and appropriate for the hypothesis being tested?

-Is the sample size sufficient to ensure adequate power to address the hypothesis being tested?

-Were correct statistical analysis used to support conclusions?

-Are there concerns about ethical or regulatory requirements being met?

Reviewer #1: Methodology is fine, but the genome data are not deposited in a repository and thus are not publicly available.

Reviewer #2: Methods are appropriate; please see comments under "Summary and General Comments"

**Results**

-Does the analysis presented match the analysis plan?

-Are the results clearly and completely presented?

-Are the figures (Tables, Images) of sufficient quality for clarity?

Reviewer #1: No observtions

Reviewer #2: Results are appropriately presented; please see comments under "Summary and General Comments"

**Conclusions**

-Are the conclusions supported by the data presented?

-Are the limitations of analysis clearly described?

-Do the authors discuss how these data can be helpful to advance our understanding of the topic under study?

-Is public health relevance addressed?

Reviewer #1: No, they are not

Reviewer #2: Conclusions are appropriately presented; please see comments under "Summary and General Comments"

**Editorial and Data Presentation Modifications?**

Reviewer #1: (No Response)

Reviewer #2: Please see comments under "Summary and General Comments"

**Summary and General Comments**

Reviewer #1: The manuscript analyses the impact of a particular gene duplication on itraconazole resistance in Madurella fahalii. This species has limited molecular tools for analysis; so, the authors selected heterologous complementation to characterize this gene.

The starting point of this work was genome sequencing of a member of this species, which is barely mentioned in the manuscript and the authors did not make justice to their own effort. One concern related to the gene duplication phenomenum is whehther this is a strain-specific observation or it is a genetic feature bradly found in this species. The authors are encouraged to assess the presence of these two copies in other isolates of this species. In addition, no effort was shown to assambly contigns and establish loci localization, which is esential to analysis gene syntheny.

Related with the expression assays, there is no evidence nor reference supporting the use of the ACT1 gene to normalize gene expression assays. Actin is a highly dinamic gene and may not of stable, and constant expression in this species. In addition, amplification efficiency of primer pairs used in RT-qPCR should be inclued in as part fo the manuscript.

It is strange that sensitivity assays were not performed following conventional assays in RPMI broth. It is known that growing fungal cells in rich media affect the susceptibility profiles. For both, the complementation assays and the expression analysis, it should be relevant to present the result generated with the other azoles tested. Moreover, a control with a non-azole antifungal drug should also be included in the analysis.

The most relevant issue, the evidence is not solid enough to support the role of this gene copy in itraconazole resistance. Actually, results suggest a modest upregulation in presence of the azole, and the complementation assays showed partial results. A stronger data set is required to link this gene with the itraconazole resistance in Madurella fahalii.

Reviewer #2: Review of PNTD-D-24-01441, “Itraconazole Resistance in Madurella fahalii Linked to a Distinct Homolog of the Gene Encoding Cytochrome P450 14-α Sterol Demethylase (CYP51)”

Authors: Isato Yoshioka, Ahmed Hassan Fahal, Satoshi Kaneko, Takashi Yaguchi

Summary: In this manuscript, the authors present data obtained from the high-quality draft genome sequence of Madurella fahalii IFM 68171 which indicates the presence of two copies of the gene encoding cytochrome P450 14-α sterol demethylase (CYP51), the target enzyme of itraconazole. These two cytochrome P450 gene sequences include a gene conserved among Madurella species (Mfcyp51A1) and a M. fahalii-specific gene 40 (Mfcyp51A2). The authors found that both genes are actively transcribed in M. fahalii and are upregulated in response to itraconazole. Furthermore, they observed that heterologous expression of each gene in Saccharomyces cerevisiae demoenstrated that transformants carrying the Mfcyp51A2 gene exhibited reduced susceptibility to itraconazole compared to those with Mfcyp51A1.

Review:

Major:

1) While this is a well written manuscript describing a novel finding of potential clinical significance, my major concern is that the Mfcyp51A2 gene was only identified in a single strain (maybe two?) of M. fahalii. It is possible that this is a rather anomalous finding and the results presented in this manuscript are not extrapolatable to other M. fahalii strains or clinical variants. Given that the authors know the sequence for the Mfcyp51A2 gene, it should be a simple matter to design specific primers for PCR amplification to test other strains to determine if this gene is common to clinical isolates of M. fahalii. This additional info would greatly enhance the impact of the manuscript.

2) Given the reduced sequence homology between Mfcyp51A1 and Mfcyp51A2 (~70%), the question remains if the enzyme is more inherently resistance to azole inhibition due to differences in the ligand binding site, or if it is simply due to increased expression of the second enzyme. The authors briefly touched on this issue in the Discussion, but it would improve the manuscript for them to comment on this aspect of the study further in their discussion and provide any additional data to support their supposition, should they indeed have it.

3) Since crystal structures are available for fungal CYP51, it could be useful and help to improve the impact of the manuscript, to construct AlphaFold homology models for Mfcyp51A1 and A2 for docking of the azole inhibitors and comparison of the two structures. This would significantly help to round out the manuscript.

4) Figure 1 is quite blurry, at least in my copy of the manuscript. If this figure is to be included in the final manuscript, please be sure that it has significant resolution to be legible.

Minor:

1) In the Materials and Methods, the authors state: “The E. coli strain was cultured in LB medium with the addition of 100 mg/L of ampicillin when necessary” please define the term “when necessary” as this is inherently ambiguous

2) Please check reference format for reference number 39. I could not find the name of the journal in this citation.

3) At four figures, the manuscript is a bit data light for a research manuscript; more indicative of a communication (see point #3 above).

PLOS authors have the option to publish the peer review history of their article (what does this mean? ). If published, this will include your full peer review and any attached files.

**Do you want your identity to be public for this peer review?** For information about this choice, including consent withdrawal, please see our Privacy Policy .

Reviewer #1: **Yes: ** Héctor M. Mora-Montes

Reviewer #2: **Yes: ** Jed N Lampe

**Figure resubmission:**
---

## [Decision Letter · Decision Letter 1]

17 Feb 2025

Dear Dr. Yaguchi,

We are pleased to inform you that your manuscript 'Itraconazole Resistance in Madurella fahalii Linked to a Distinct Homolog of the Gene Encoding Cytochrome P450 14-α Sterol Demethylase (CYP51)' has been provisionally accepted for publication in PLOS Neglected Tropical Diseases.

Best regards,

Max Carlos Ramírez-Soto, BSc, MPH, PhD, FRSPH

Academic Editor

Marcio Rodrigues

Section Editor

Shaden Kamhawi

co-Editor-in-Chief

Paul Brindley

co-Editor-in-Chief

None

Reviewer's Responses to Questions

**Key Review Criteria Required for Acceptance?**

**Methods**

-Are the objectives of the study clearly articulated with a clear testable hypothesis stated?

-Is the study design appropriate to address the stated objectives?

-Is the population clearly described and appropriate for the hypothesis being tested?

-Is the sample size sufficient to ensure adequate power to address the hypothesis being tested?

-Were correct statistical analysis used to support conclusions?

-Are there concerns about ethical or regulatory requirements being met?

Reviewer #1: The authors properly addressed my concerns. The manuscript is now ready for publication.

Reviewer #2: Acceptable

**Results**

-Does the analysis presented match the analysis plan?

-Are the results clearly and completely presented?

-Are the figures (Tables, Images) of sufficient quality for clarity?

Reviewer #1: The authors properly addressed my concerns. The manuscript is now ready for publication.

Reviewer #2: Acceptable

**Conclusions**

-Are the conclusions supported by the data presented?

-Are the limitations of analysis clearly described?

-Do the authors discuss how these data can be helpful to advance our understanding of the topic under study?

-Is public health relevance addressed?

Reviewer #1: The authors properly addressed my concerns. The manuscript is now ready for publication.

Reviewer #2: Acceptable

**Editorial and Data Presentation Modifications?**

Reviewer #1: (No Response)

Reviewer #2: Acceptable

**Summary and General Comments**

Reviewer #1: The authors properly addressed my concerns. The manuscript is now ready for publication.

Reviewer #2: Acceptable

PLOS authors have the option to publish the peer review history of their article (what does this mean? ). If published, this will include your full peer review and any attached files.

**Do you want your identity to be public for this peer review?** For information about this choice, including consent withdrawal, please see our Privacy Policy .

Reviewer #1: **Yes: ** Héctor M. Mora-Montes

Reviewer #2: No

---

## [Editor Report · Acceptance letter]

Dear Dr. Yaguchi,

We are delighted to inform you that your manuscript, "Itraconazole Resistance in Madurella fahalii Linked to a Distinct Homolog of the Gene Encoding Cytochrome P450 14-α Sterol Demethylase (CYP51)," has been formally accepted for publication in PLOS Neglected Tropical Diseases.

Best regards,

Shaden Kamhawi

co-Editor-in-Chief

Paul Brindley

co-Editor-in-Chief
